# ACUTUM: WHEN GENERALIZATION MEETS ADAPTABILITY

## ABSTRACT

In spite of the slow convergence, stochastic gradient descent (SGD) is still the most widely used optimization method due to its outstanding generalization ability and simplicity. On the other hand, adaptive methods have attracted much more attention of optimization and machine learning communities, both for the leverage of life-long information and for the deep and fundamental mathematical theory. Taking the best of both worlds is the most exciting and challenging question in the field of optimization for machine learning. In this paper, we revisit existing adaptive gradient methods from a novel point of view, which reveals a fresh understanding of momentum. Our new intuition empowers us to remove the second moment in Adam without the loss of performance. Based on our view, we propose a novel optimization method, the *acute adaptive momentum* (Acutum). To the best of our knowledge, Acutum is the first adaptive gradient method without second moments. Experimentally, we demonstrate that our method has a faster convergence rate than Adam/Amsgrad, and generalizes as well as SGD with momentum. We also provide a convergence analysis of our proposed method to complement our intuition.

## 1 INTRODUCTION

With the rapid development of neural network architectures (Goodfellow et al., 2016), training algorithms have attracted much more attention in the modern machine learning community. Due to the network size and data amount increasing dramatically, calculating the full gradient of data and implementing the full gradient descent (GD) become computationally expensive. Therefore, stochastic gradient descent (SGD) (Robbins & Monro, 1951) becomes the most practical optimization method for training deep neural networks (DNNs). In each iteration, SGD samples mini-batch data and computes the gradient corresponding to the mini-batch. Although SGD is computationally affordable, it needs a mechanism of fine-tune the learning rate, e.g., linear decay or exponential decay, to converge efficiently. In fact, it is pretty brittle to tune the learning rate dynamically with SGD.

To release the learning rate tuning burden of SGD and accelerate its convergence, several adaptive variants of SGD were proposed, including Adagrad (Duchi et al., 2011), Adadelta (Zeiler, 2012), RMSProp (Hinton et al., 2012), Adam (Kingma & Ba, 2014), etc. The descent direction is element-wise automatically adapted by the first moment and the second moment with an exponential average of gradients. Such a descent direction has achieved great improvements in practice, while the fundamental interpretation is still unclear. Besides, the theoretical regret analysis of these online learning algorithms has become completed gradually in convex objective settings. Due to the fast decay of exponential moving average, Adam cannot converge even in simple convex cases as shown in (Reddi et al., 2019). Amsgrad (Reddi et al., 2019) addressed this issue by keeping an extra non-decreasing sequence to buffer the fast decay.

On the other hand, although adaptive methods are easy to use and fast to converge, the generalization results cannot be as good as SGD (Wilson et al., 2017). Various works (Chen & Gu, 2018; Luo et al., 2019) have been proposed to make algorithms not only converge faster but also has good generalization. However, these algorithms still inherit the second moment following Adam. Different from existing works, we provide a novel view for this problem by revisiting adaptive algorithms Through the view of *direction angle* between the descent direction and the momentum, we are able to demonstrate a fresh understanding of momentum.

In summary, we propose a new algorithm, called **acu**te adaptive momen**tum** (Acutum), to find adaptive gradients with only the first moment. To remove the second moments safely, we investigate their properties by reformulating the proximal subproblem of Adagrad. The main observation is that second moments essentially penalize the projection of the current descent direction on the exponential moving average of previous gradients. The new intuition allows us to remove second moments in Adam without any loss of performance. To the best of our knowledge, it is the first time to remove the second moments for adaptive gradient methods. As shown by our experiment, the proposed algorithm (Acutum) outperforms Adam/Amsgrad in wall-clock time while almost preserving the generalization ability of SGD.

In particular, we make the following contributions:

- We propose a new optimization algorithm called Acutum, which doesn't need to compute the second moments. To the best of our knowledge, it is the first time to obtain the adaptive gradients with only the first moments. The intuition behind our algorithm is to remove second moments by attaching their purpose to the first moment through Adagrad's proximal subproblem inspiration.

- We provide a regret convergence analysis of Acutum on the convex setting, based on the analysis of (Kingma & Ba, 2014; Reddi et al., 2019; Chen & Gu, 2018), and prove a data-dependent regret bound, complementing our intuition.

- We also provide various experiments about our proposed Acutum on training modern deep models. The experimental results empirically show that Acutum takes the best from both Adam-type methods and SGD.

**Notations.** In the rest of this paper, we assume any decision variables $\boldsymbol{\theta} \in \mathbb{R}^d$ where $d$ denotes the dimension of parameters. We denote the $l_2$ norm of a given vector $\boldsymbol{\theta}$ by $\|\boldsymbol{\theta}\| = \sqrt{\sum_{i=1}^d \theta_i^2}$. With slightly abuse of notation, we write arithmetic symbols as element-wise operations for vectors, e.g., $\mathbf{a}^2 = [a_1^2, a_2^2, \ldots]^T, \mathbf{a}/\mathbf{b} = [a_1/b_1, a_2/b_2, \ldots]^T$. We denote by $\lfloor x \rceil$ the greatest integer less than or equal to the real number $x$. Given any integers $x, y$, where $y \neq 0$, we denote by $x \pmod y$ the remainder of the Euclidean division of $x$ by $y$. In the optimization setting, we denote by $f_i(\cdot)$ the loss function when we feed the $i$-th training data into our model, we also denote $f_S(\cdot)$ when we feed a collection of samples, i.e., $f_S(\cdot) := \sum_{i \in S} f_i(\cdot)$. Given any point $\boldsymbol{\theta} \in \mathbb{R}^d$ and any convex set $\mathcal{X} \subseteq \mathbb{R}^d$, we denote by $\Pi_{\mathcal{X}}(\boldsymbol{\theta})$ the *projection* of $\boldsymbol{\theta}$ onto $\mathcal{X}$, i.e., $\arg\min_{\boldsymbol{\mu} \in \mathcal{X}} \|\boldsymbol{\mu} - \boldsymbol{\theta}\|$.

## 2 RELATED WORK

The increasing size of deep neural networks and the amount of train data have dramatically induced the difficulty of training neural networks. The huge parameter size makes the landscape of neural networks more non-convex, and the amount of train data requires more computation resources to calculate the gradient. Therefore, all the facts urge researchers to design faster optimization algorithms for deep neural networks, while maintaining a reasonable generalization ability.

SGD-momentum is widely used in training large-scale neural networks while the excellent generalization ability of SGD is vulnerable to its learning rate. Meanwhile, researchers began to focus on the design of adaptive gradient methods for a fast and simple optimizer. Adagrad proposed in (Duchi et al., 2011) introduced the second moment to obtain a self-adaptive learning rate, and hence freed researchers' hands of parameter tuning. The update rules of Adagrad can be formulated as follows:

$$\boldsymbol{\theta}_{t+1} = \boldsymbol{\theta}_t - \alpha_t \frac{\mathbf{g}_t}{\sqrt{\mathbf{v}_t}},$$

where $\mathbf{g}_t$ denotes the stochastic gradient, $\mathbf{v}_t$ is the accumulation of gradient's second moments, i.e., $\mathbf{v}_t = \frac{1}{t} \sum_{\tau=1}^t \mathbf{g}_\tau^2$, and $\alpha_t$ is the decreasing learning rate with $\alpha_t = O(1/\sqrt{t})$. Theoretically, Adagrad improves the convergence of regret from $O(\sqrt{d}/\sqrt{T})$ to $O(1/\sqrt{T})$ for the convex objectives with sparse gradients. However, in practice, people realize that adaptive gradient of Adagrad, i.e., $\mathbf{g}_t/\sqrt{\mathbf{v}_t}$ goes to zero very quickly due to $\mathbf{v}_t$ accumulating to large number quickly as the algorithm proceeds. To make it through, RMSProp (Hinton et al., 2012) introduced the exponential decay in the second moment to control the accumulation speed of second moment in Adagrad. Furthermore,

(Kingma & Ba, 2014) incorporated the momentum into RMSProp and proposed Adam. The detailed procedure of Adam can be formulated as:

$$\boldsymbol{\theta}_{t+1} = \boldsymbol{\theta}_t - \alpha_t \frac{\mathbf{m}_t}{\sqrt{\mathbf{v}_t}}, \tag{1}$$

where $\mathbf{m}_t$ is the exponential decay of momentum, i.e., $\mathbf{m}_t = \sum_{\tau=1}^t (1-\beta_1)\beta_1^{t-\tau}\mathbf{g}_\tau$, and $\mathbf{v}_t$ is the exponential decay of second moment, i.e., $\mathbf{v}_t = \sum_{\tau=1}^t (1-\beta_2)\beta_2^{t-\tau}\mathbf{g}_\tau^2$.

The faster convergence, robust hyper-parameters and good performance on CV and NLP's tasks make Adam become one of the most successful optimizers these years. However, (Reddi et al., 2019) showed Adam does not converge in certain convex cases, and proposed Amsgrad to correct the direction of Adam with imposing an increasing sequence of the second moment.

Although convergence, in theory, is fixed and the training speed is still fast, the generalization ability of adaptive algorithms is still worse than SGD with momentum. Thus, there are a lot of studies to improve the generalization performance of Adam types algorithms. Almost all of these works try to make some connections between Adam and SGD-momentum such that the proposed algorithm converges faster like Adam while generalizing better as SGD. For example, (Zhang et al., 2017) proposed a normalized direction-preserving Adam (ND-Adam). To make the direction of the gradients preserved, the authors adjusted the learning rate to each weight vector as a whole, instead of each individual weight. It could be thought as an adjustment of learning rate dynamically in each iteration, instead of the learning rate decay. (Loshchilov & Hutter, 2017) proposed to modify Adam to decay all weights the same regardless of the gradient variances. (Keskar & Socher, 2017) proposed to improve the generalization performance by investigating a hybrid strategy that begins training with an adaptive method and switches to SGD when appropriate. The experimental results show that it obtains a better generalization with the convergence speed sacrificing. With a similar insight, (Luo et al., 2019) introduced the element-wise clipping technique to the second moment and proposed AdaBound, which allows the algorithm behavior switched from a clipped Adam to SGD-momentum as the algorithm proceeds. Besides, (Chen & Gu, 2018) proposed Padam which tried to find the balanced state between Adam and SGD-momentum through tuning the order of the second moment in the update paradigm.

All the aforementioned methods try to find the connection between Adam and SGD-momentum, thus all these algorithms take the second moment adaptation as a grant. The difference between previous works and ours is that we suggest to remove the second moment and propose an angle based algorithm, by revisiting the original idea of second moment adaptation. Such intuition could be of independent interest in the field of the optimization community.

## 3 THE ACUTE ADAPTIVE MOMENTUM ALGORITHM

In this section, we first revisit existing adaptive algorithms from a novel point of view, then introduce the intuition and provide the algorithm procedure of our method Acutum. The performance guarantees established in Section 4 and evaluation results in Section 5 validate our insights.

From the above section, one may notice that almost all the adaptive methods utilize second moments to adjust the individual magnitude of the updates. However, the efficient calculation of second moments, see Equation (1), is only proposed in Adagrad (Duchi et al., 2011). Let's take a look at the original update paradigm of Adagrad:

$$\boldsymbol{\theta}_{t+1} = \boldsymbol{\theta}_t - \alpha_t \mathbf{G}_t^{-1/2}\mathbf{g}_t, \tag{2}$$

where $\mathbf{G}_t = \sum_{\tau=1}^t \mathbf{g}_\tau \mathbf{g}_\tau^T$, and $\mathbf{g}_\tau$ denotes the stochastic gradient calculated at iteration $\tau$. Recall that in each iteration, SGD seeks the optimal solution of a subproblem constructed from a quadratic approximation of the objective. Similarly, we can write the update procedure of Adagrad in Equation (2) as

$$\boldsymbol{\theta}_{t+1} = \underset{\boldsymbol{\theta}}{\arg\min}\ \underbrace{(\boldsymbol{\theta}-\boldsymbol{\theta}_t)^T\mathbf{g}_t}_{T_1} + \underbrace{\frac{1}{2\alpha_t}(\boldsymbol{\theta}-\boldsymbol{\theta}_t)^T\mathbf{G}_t^{1/2}(\boldsymbol{\theta}-\boldsymbol{\theta}_t)}_{T_2} \tag{3}$$

where the first-order penalty $T_1$ leads the parameter $\boldsymbol{\theta}_t$ to the opposite direction to the current gradient $\mathbf{g}_t$. Note that Adagrad can adaptively adjust each individual dimension of $\boldsymbol{\theta}_t$ through introducing

$\mathbf{G}_t$ in the quadratic term $T_2$, while SGD cannot. If we substitute the definition of $\mathbf{G}_t$ back into Equation (3), $T_2$ can be formulated as:

$$T_2 = \frac{1}{2\alpha_t} (\boldsymbol{\theta} - \boldsymbol{\theta}_t)^T \mathbf{G}_t^{1/2} (\boldsymbol{\theta} - \boldsymbol{\theta}_t) = \frac{1}{2\alpha_t} (\boldsymbol{\theta} - \boldsymbol{\theta}_t)^T \left( \sum_{\tau=1}^t \mathbf{g}_\tau \mathbf{g}_\tau^T \right)^{1/2} (\boldsymbol{\theta} - \boldsymbol{\theta}_t). \qquad (4)$$

By Young's inequality (Young, 1912), we can obtain an upper bound of $T_2$ as follows:

$$T_2 \le \underbrace{\frac{1}{8\alpha_t} \|\boldsymbol{\theta} - \boldsymbol{\theta}_t\|^2}_{T_3} + \underbrace{\frac{1}{2\alpha_t} \sum_{\tau=1}^t \left\| (\boldsymbol{\theta} - \boldsymbol{\theta}_t)^T \mathbf{g}_\tau \right\|^2}_{T_4}. \qquad (5)$$

Combining Equation (5) and Equation (3), we can approximate the update rule of Adagrad as follows:

$$\boldsymbol{\theta}_{t+1} \approx \arg\min_{\boldsymbol{\theta}} \underbrace{(\boldsymbol{\theta} - \boldsymbol{\theta}_t)^T \mathbf{g}_t}_{T_1} + \underbrace{\frac{1}{8\alpha_t} \|\boldsymbol{\theta} - \boldsymbol{\theta}_t\|^2}_{T_3} + \underbrace{\frac{1}{2\alpha_t} \sum_{\tau=1}^t \left\| (\boldsymbol{\theta} - \boldsymbol{\theta}_t)^T \mathbf{g}_\tau \right\|^2}_{T_4}.$$

We can think the upper bound of the Adagrad subproblem as the combination of the standard first and second order penalty as $T_1$ and $T_3$, with a penalty for the projections of current descent $\boldsymbol{\theta} - \boldsymbol{\theta}_t$ on the previous gradients $\mathbf{g}_\tau$, i.e., $T_4$.

With these observations in mind, a natural question comes out: how can we understand $T_4$ in neural network training tasks or in general online learning tasks?

Before answering this question, we first describe the practical usage of training samples. The training procedure of DNNs can be stated as follows. The algorithm randomly shuffles the whole training dataset $\mathbb{A}$ and partitions $\mathbb{A}$ into mini-batch of equal size $\{\mathbb{A}_0, \mathbb{A}_1, \ldots, \mathbb{A}_{p-1}\}$, then the algorithm is fed with the sample in a fixed order, say, first $\mathbb{A}_0$, then $\mathbb{A}_1$, and so on. The whole procedure repeats after each pass of the whole dataset.

To make our description more clear, we take the first pass as an example. Let $\nabla f_{\mathbb{A}_t}(\boldsymbol{\theta}_t)$ be the gradient calculated on iteration $t$ by using the sample in subset $\mathbb{A}_t$, where $0 \le t < p$ due to the first pass. Note that the loss function is the sum of whole samples, e.g., $\frac{1}{n} \sum_{i=1}^n f_i(\boldsymbol{\theta})$, if we utilize $\nabla f_{\mathbb{A}_i}(\boldsymbol{\theta}_t)$ to directly update parameters like SGD, e.g., $\boldsymbol{\theta}_{t+1} = \boldsymbol{\theta}_t - \alpha \nabla f_{\mathbb{A}_t}(\boldsymbol{\theta}_t)$, the batch loss $\sum_{i \in \mathbb{A}_t} f_i(\boldsymbol{\theta})$ will decrease since it aligns with the opposite direction of its gradient, e.g., $(\boldsymbol{\theta}_{t+1} - \boldsymbol{\theta}_t)^T \nabla f_{\mathbb{A}_t}(\boldsymbol{\theta}_t) = -\|\nabla f_{\mathbb{A}_t}(\boldsymbol{\theta}_t)\|^2 < 0$. However, for loss corresponding to the sample not in $\mathbb{A}_t$, it is highly possible that $(\boldsymbol{\theta}_{t+1} - \boldsymbol{\theta}_t)^T \nabla f_{\mathbb{A}_i}(\boldsymbol{\theta}_t) > 0$. On other words, only using $-\nabla f_{\mathbb{A}_i}(\boldsymbol{\theta}_t)$ as update direction will decrease the loss function corresponding to $\mathbb{A}_t$ but increase the total loss function except $\mathbb{A}_t$.

Ideally, if there exists a direction $\boldsymbol{\theta}_{t+1} - \boldsymbol{\theta}_t$ which is orthogonal to $\nabla f_{\mathbb{A}_i}(\boldsymbol{\theta}_t)$ for any $i \ne t \pmod p$, and forms an acute angle with $\nabla f_{\mathbb{A}_t}(\boldsymbol{\theta}_t)$, then $\theta_{t+1}$ can guarantee a sufficient descent for the loss function corresponding to $\mathbb{A}_t$ while not increase the loss function outside $\mathbb{A}_t$.

Then a natural update of $\boldsymbol{\theta}_{t+1}$ to achieve the above goal can be obtained by solving the following sub-problem:

$$\arg\min_{\boldsymbol{\theta}} \quad c_2 \|\boldsymbol{\theta} - \boldsymbol{\theta}_t\|^2 + c_1 (\boldsymbol{\theta} - \boldsymbol{\theta}_t)^T \mathbf{g}_t$$
$$s.t. \quad (\boldsymbol{\theta} - \boldsymbol{\theta}_t)^T \nabla f_{\mathbb{A}_i}(\boldsymbol{\theta}_t) = 0, i \ne t \pmod p, \qquad (6)$$

In practice, however, the computational cost of computing $\nabla f_{\mathbb{A}_i}(\boldsymbol{\theta}_t)$ for all $i$ is expensive in each iteration. As a natural solution, we can approximate $\nabla f_{\mathbb{A}_i}(\boldsymbol{\theta}_t)$ with previous gradients $f_{\mathbb{A}_i}(\boldsymbol{\theta}_i)$, which is exactly $\mathbf{g}_i$ calculated in Adagrad in iteration $i$ before iteration $t$, where $i \ne t \pmod p$. Hence a soft margin variant of Equation (6) can be formulated as

$$\boldsymbol{\theta}_{t+1} = \arg\min_{\boldsymbol{\theta}} c_{2,0} \|\boldsymbol{\theta} - \boldsymbol{\theta}_t\|^2 + c_{2,1} \sum_{\tau=1}^t \left\| (\boldsymbol{\theta} - \boldsymbol{\theta}_t)^T \mathbf{g}_\tau \right\|^2 + c_1 (\boldsymbol{\theta} - \boldsymbol{\theta}_t)^T \mathbf{g}_t, \qquad (7)$$

which is exactly the formulation in Equation (5) and Equation (3)! Thus, one can reasonably consider that *Adagrad decreases the cumulative loss by guaranteeing a descent on the current batch loss while does not increase previous batch loss.*

Although the approximation of $\nabla f_{\mathbb{A}_i}(\boldsymbol{\theta}_t)$ using previous gradients $\mathbf{g}_i$ in Adagrad performs well in practice, there is still a large accuracy gap. For example, let $\mathbf{g}_{t-1}$ and $\mathbf{g}_{t-2}$ denote the gradients $\nabla f_{\mathbb{A}_{i-1}}(\boldsymbol{\theta}_{t-1})$ and $\nabla f_{\mathbb{A}_{i-2}}(\boldsymbol{\theta}_{t-2})$, respectively. It is often the case that the approximation error between $\nabla f_{\mathbb{A}_{i-1}}(\boldsymbol{\theta}_t)$ and $\mathbf{g}_{t-1}$ is smaller than that between $\nabla f_{\mathbb{A}_{i-2}}(\boldsymbol{\theta}_t)$ and $\mathbf{g}_{t-2}$, assuming the objective is local smoothness[1]. In other words, the approximation accuracy of $\mathbf{g}_\tau$ to $\nabla f_{\mathbb{A}_i}(\boldsymbol{\theta}_t)$ decays as $\tau$ decreases. Inspired by this observation, we substitute the arithmetic average in Equation (7) to exponential moving average and reassign the coefficients of $\mathbf{g}_\tau$ like Adam/momentum methods, then Equation (7) becomes

$$\boldsymbol{\theta}_{t+1} = \arg\min_{\boldsymbol{\theta}} \; c_{2,0} \|\boldsymbol{\theta} - \boldsymbol{\theta}_t\|^2 + c_{2,1} \sum_{\tau=1}^{t} \left\| (1-\beta)\,\beta^{t-\tau}\,(\boldsymbol{\theta} - \boldsymbol{\theta}_t)^T \mathbf{g}_\tau \right\|^2 + c_1 \,(\boldsymbol{\theta} - \boldsymbol{\theta}_t)^T \mathbf{g}_t.$$

This is what RMSProp actually does. Therefore, *we obtain the same optimization intuition of RMSProp as Adagrad but add weight of the approximate gradient according to its approximation accuracy.*

Enlightened by the above analysis, we are ready to propose a new adaptive momentum method. Let $\mathbf{m}_t$ denote the approximated gradient of the previous batch loss at current parameters. With the following observation: (i) if $\boldsymbol{\theta}_t - \boldsymbol{\theta}_{t+1}$ forms acute angles with both $\mathbf{g}_t$ and $\mathbf{m}_t$, rather than penalizing the projection like what Adagrad and RMSProp do, we can obtain both descent on current and previous batches; (ii) to handle the case when the estimation of previous gradients using $\mathbf{m}_t$ is not accurate, we expect the current gradient $\mathbf{g}_t$ to dominate the descent direction, which is denoted as $\hat{\mathbf{m}}_t$. It can be guaranteed by requiring the projection length of $\hat{\mathbf{m}}_t$ on $\mathbf{g}_t$ is less than the length of $\mathbf{g}_t$ if properly regularizing $\mathbf{m}_t$. Then, we propose a new iterative subproblem in the following:

$$\boldsymbol{\theta}_{t+1} = \arg\min_{\boldsymbol{\theta}} \; c_{2,0} \|\boldsymbol{\theta} - \boldsymbol{\theta}_t\|^2 + c_1 \,(\boldsymbol{\theta} - \boldsymbol{\theta}_t)^T \left( \mathbf{g}_t + \|\mathbf{g}_t\| \frac{\hat{\mathbf{m}}_t}{\|\hat{\mathbf{m}}_t\|} \right),$$

$$\text{where } \hat{\mathbf{m}}_{t+1} = \mathbf{g}_t + \|\mathbf{g}_t\| \frac{\hat{\mathbf{m}}_t}{\|\hat{\mathbf{m}}_t\|}. \tag{8}$$

With such formulation, the following properties hold:

- We ensure the above two observations, specifically, with the update of momentum $\hat{\mathbf{m}}_t$, we can guarantee that $\|\hat{\mathbf{m}}_{t+1}\| = O(\|\mathbf{g}_t\|)$.
- We attach the properties of the second moment to the first-order operation, inspired by Adagrad proximal subproblem.

The proposed algorithm Acutum is summarized in Algorithm 1.

---

**Algorithm 1** Acute Adaptive Momentum method (Acutum)

1: **Input:** initial point $\boldsymbol{\theta}_0 \in \mathcal{X}$; step size $\{\alpha_t\}$, momentum parameters $\{\beta_t\}$
2: set $\hat{\mathbf{m}}_0 = 0$
3: **for** $t = 1$ to $T$ **do**
4:     $\mathbf{g}_t = \nabla f_t(\boldsymbol{\theta}_t)$
5:     $\hat{\mathbf{m}}_t = \mathbf{g}_t + \beta_{1,t} \|\mathbf{g}_t\| / (\|\hat{\mathbf{m}}_{t-1}\| + \epsilon) \cdot \hat{\mathbf{m}}_{t-1}$
6:     $\boldsymbol{\theta}_{t+1} = \Pi_{\mathcal{X}} (\boldsymbol{\theta}_t - \alpha_t \cdot \hat{\mathbf{m}}_t)$
7: **end for**
8: **Return:** $x_{T+1}$

---

Note that, at step 5 we use $\epsilon$ to prevent precision overflow caused by a tiny $\|\hat{\mathbf{m}}_{t-1}\|$. $\beta_{1,t}$ is the momentum weight. We specify its value in the theoretical analysis in Section 4. In practice, we can simply set $\beta_{1,t}$ to be 1. Compared with other adaptive gradient methods, Acutum has fewer hyper-parameters, and is much easier to tune deep models. Besides, it is also computation-friendly (for both time and space complexities) since the algorithm does not need second moments.

---

[1]The local smoothness assumption is widely used in the convergence analysis for convex and non-convex objectives, see (Nesterov, 1998; Johnson & Zhang, 2013).

## 4 CONVERGENCE ANALYSIS

In this section, we analyze the convergence rate of our proposed algorithm Acutum in an online convex setting, i.e., we assume the convexity of a sequence of loss functions $f_1, f_2, \ldots, f_T$, as shown in the following:

**Assumption 4.1** (Convexity Assumption). *For each loss function $f_t(\boldsymbol{\theta})$ where $1 \leq t \leq T$, we assume it is convex. That is, for any $\mathbf{x}, \mathbf{y} \in \mathcal{X}$, we have*

$$f_t(\mathbf{y}) \geq f_t(\mathbf{x}) + \nabla f_t(\mathbf{x})^T(\mathbf{y} - \mathbf{x}),$$

where $\mathcal{X}$ is the feasible set of function $f_1, f_2, \ldots, f_T$.

Notice that this assumption is widely used for theoretical analysis, in both stochastic optimization methods (Johnson & Zhang, 2013; Defazio et al., 2014; Shalev-Shwartz & Zhang, 2013) and online learning methods (Duchi et al., 2011; Kingma & Ba, 2014; Reddi et al., 2019). Rather than consider the objective as a finite-sum function, here we analyze the regret from a online learning perspective. That is to say, at each time step $t$, we are given the loss $f(\boldsymbol{\theta}_t)$ after selecting some feasible solution $\boldsymbol{\theta}_t$. Then we continue to pick the next point $\boldsymbol{\theta}_{t+1}$ based on the previous losses.

We adopt *online regret* as the convergence metric, which equals to the relative difference between the algorithm's cumulative loss and the offline minimum value. The definition of regret is straight-forward, i.e.,

$$R(T) := \sum_{t=1}^{T} f_t(\boldsymbol{\theta}_t) - \min_{\boldsymbol{\theta} \in \mathcal{X}} \sum_{t=1}^{T} f_t(\boldsymbol{\theta}),$$

where $\mathcal{X}$ is the feasible set over all steps. Our purpose is to find a sequence of points $\{\boldsymbol{\theta}_t\}_{1 \leq t \leq T}$ to minimize the overall regret $R(T)$. We state the main theoretical results in the following theorem.

**Theorem 4.2.** *Suppose Assumption 4.1 holds. We set $\epsilon > 0$, $\alpha_t = \alpha\sqrt{d/t}$, $\beta_{1,t} = \beta\lambda^{t-1}$, where $\beta, \lambda \in (0, 1)$. If the diameter of the convex feasible set $\mathcal{X}$ are bounded, i.e., $\|\boldsymbol{\theta} - \boldsymbol{\theta}^*\|_\infty \leq D_\infty$ for all $\boldsymbol{\theta} \in \mathcal{X}$, and $f_t$ has bounded gradients, i.e., $\|\nabla f_t(\boldsymbol{\theta})\|_\infty \leq G_\infty$ for all $\boldsymbol{\theta} \in \mathcal{X}$, $1 \leq t \leq T$, then the regret of Algorithm 1 satisfies:*

$$R(T) \leq \frac{D_\infty^2}{2\alpha}\sqrt{dT} + 3G_\infty\alpha\sqrt{(1 + \log T)\, d} \sum_{i=1}^{d} \|g_{1:T,i}\|_2 + \frac{\sqrt{d}\beta D_\infty^2}{2\alpha\left(1 - \lambda\right)^2}. \tag{9}$$

Theorem 4.2 shows that our proposed algorithm has a similar order of step size to online gradient descent, i.e., $O(\sqrt{d/t})$, because of removing the second moments. In this situation, weight decay and appropriate learning rate decay strategies have much more influence on Acutum than Adam-type optimizers. Also, from the proof of Theorem 4.2 in appendix A.2, we find that the same order of regret bound can also be obtained even with a more modest momentum decay $\beta_{1,t} = \beta/t$. Furthermore, we show in the following corollary that our Acutum enjoys a regret bound of $\tilde{O}(\sqrt{T})$, which is comparable to the best of known bound for general convex online learning problems.

**Corollary 4.3.** *Frame the hypotheses of Theorem 4.2, for all $T \geq 1$, the regret of Algorithm 1 satisfies $R(T) = \tilde{O}(\sqrt{T})$.*

Corollary 4.3 illustrates that Acutum obtains $R_T = o(T)$ for all situations which is independent of the sparsity of the data features. Besides, since $\lim_{T\to\infty} R_T/T \to 0$, Acutum will converge to the optimal solution when the objectives are convex.

## 5 EXPERIMENTS

In this section, we conduct extensive experiments on image classification tasks. We want to validate that Acutum with only first moments can not only obtain a better generalization performance than adaptive gradient methods, but also converges faster than SGD-momentum.

First, note that introducing weight decay is equivalent to adding $l_2$ regularization to the objectives, and has a significant impact on the generalization ability of optimizers. To better characterize generalization performance, we conduct experiments from two perspectives: (i) In Section 5.1, we record

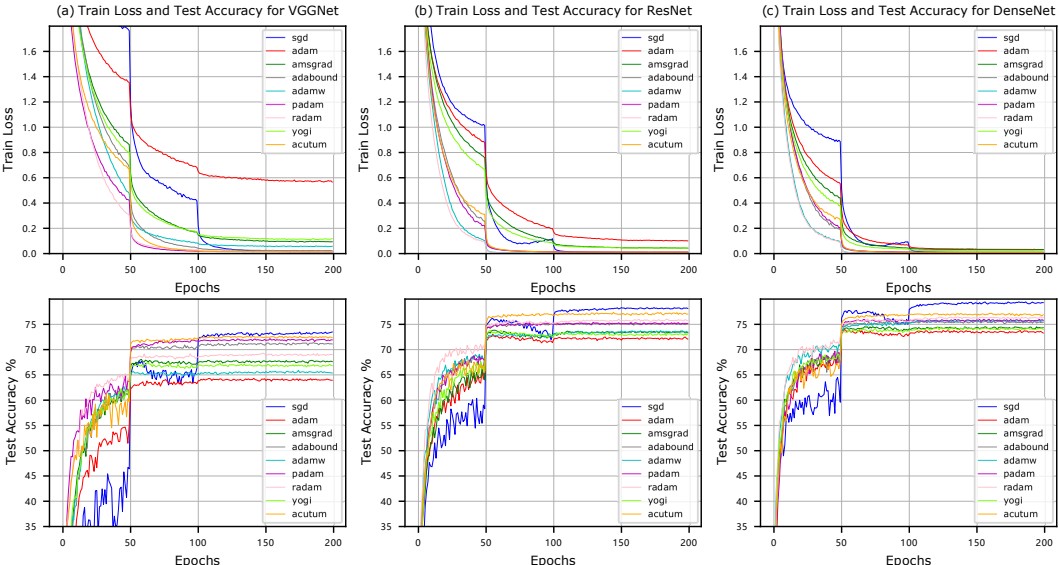

Figure 1: With a same weight decay, train loss and test accuracy of different optimizers for VGGNet, ResNet and DenseNet on CIFAR-100.

the convergence of training loss and the test accuracy of all optimizers with fixed weight decays (they optimize the same objective function). (ii) In Section 5.2, we adopt the optimal weight decays for all optimizers, then investigate their generalization ability based on the test accuracy.

Now we introduce the experimental settings of our experiments, which are general and commonly used. Specifically, we implement all of our codes in Pytorch platform version 1.0+ within Python 3.6+. To obtain sufficient comparisons with adaptive gradient variants, we choose various baselines, including SGD-momentum, Adam (Kingma & Ba, 2014), Amsgrad (Reddi et al., 2019), Adamw (Loshchilov & Hutter, 2017), Yogi (Zaheer et al., 2018), Padam (Chen & Gu, 2018), Radam (Liu et al., 2019) and Adabound (Luo et al., 2019). For the image classification tasks, we use two datasets CIFAR-10, CIFAR-100 (Krizhevsky et al., 2009), and test three different CNN architectures including VGGNet (Simonyan & Zisserman, 2014), ResNet (He et al., 2016) and DenseNet (Huang et al., 2017) . To obtain stable convergence, we run 200 epochs, and decay the learning rate by $0.1$ every $50$ epochs. We perform cross-validation to choose the best learning rates for all optimizers and second moment parameters $\beta_2$ for all adaptive gradient methods.

## 5.1 EXPERIMENT WITH FIXED WEIGHT DECAY

We first evaluate CIFAR-100 dataset. In this experiment, the values of weight decay in all optimizers are fixed, and are chosen to be the weight decay in SGD-momentum when it achieves the maximum test accuracy.

From the first row of Figure 1, i.e, the training curves of three tests, we observe that our proposed Acutum significantly outperforms Adam, and has comparable rate with Padam and Adabound when we fix the weight decay. Also, in the second row of Figure 1, i,e, the test accuracy, our Acutum outperforms all benchmarks except only SGD-momentum. However, the training loss converges much slower when implemented with SGD-momentum which is also recorded in other literature. The above evaluation results validate that our method achieves both fast convergence and good generalization when the weight decay is fixed. Note that the AdamW method reduces training loss quickly, however, there is a large generalization gap between it and other optimizers, including SGD-momentum, Acutum, Padam.

## 5.2 EXPERIMENT WITH OPTIMAL WEIGHT DECAY

In this section, we conduct experiments on CIFAR-10 dataset. Specifically, we find the optimal weight decay for each optimizer which can obtain the maximum test accuracy.

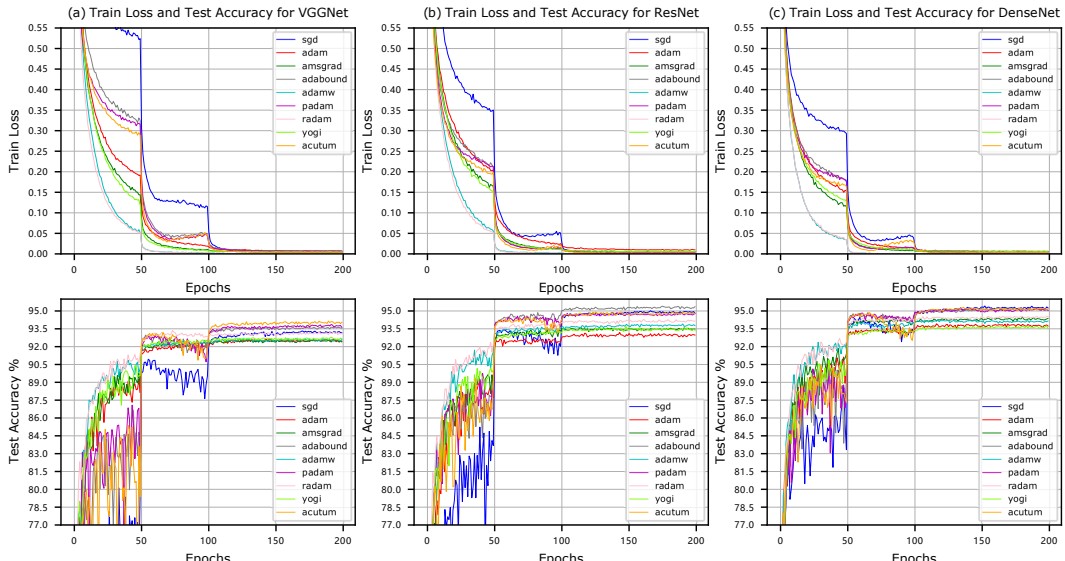

Figure 2: With different weight decays, train loss and test accuracy of different optimizers for VG-GNet, ResNet and DenseNet on CIFAR-10.

As shown in Figure 2, Adam, AdamW and Amsgrad perform worse than the other optimizers in the plots of test accuracy. In this optimal weight decay setting, Padam, Adabound and Acutum can even have better test performance than SGD-momentum on VGGNet if sacrificing some convergence rate. Besides, they outperform Adam by more than 2 percentage points for the test accuracy. The experimental results demonstrate that optimizers which converge faster than Acutum generalize worse. Similar to results in previous section, Acutum can effectively bridge the generalization gap, and has a comparable performance with the state-of-the-art Adam's variants while only requiring first moments.

## 6 CONCLUSION

In this paper, we revisited the existing adaptive optimization methods from a novel point of view. We found that the widely used second moments essentially penalize the projection of the current descent direction on the exponential moving average of previous gradients. In other words, it aims to decrease the current batch loss while does not increase previous batch loss. Following such new idea, we investigate how to obtain descent on both current and previous batches. Specifically, we proposed a new method, acute adaptive momentum (Acutum). It removes the computation-complicated second moments and constructed a decent direction by forming acute angles with both current and (approximated) previous gradients. We analyzed its convergence property in the online convex setting. The extensive evaluations demonstrate that our Acutum can effectively bridge the gap between fast convergence and good generalization, i.e., the advantages of both adaptive gradient methods and SGD-momentum.

For future work, it would be interesting to evaluate our method in other types of tasks and deep models, especially NLP and GAN problems. Besides, explaining the generalization ability of SGD is one of the most significant open problems in this field. We also hope that one could improve the convergence rate of SGD by only scheduling the learning rate carefully and dynamically.

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

# A PROOF OF THE MAIN RESULTS

## A.1 SOME IMPORTANT LEMMAS

In this section, we give several important definitions and lemmas which will be used in the proof of the regret convergence.

**Lemma A.1.** *For some convex feasible set $\mathcal{X} \subset \mathbb{R}^d$, suppose $\hat{\boldsymbol{\theta}}_1 = \min_{\boldsymbol{\theta} \in \mathcal{X}} \|\boldsymbol{\theta} - \boldsymbol{\theta}_1\|$ and $\hat{\boldsymbol{\theta}}_2 = \min_{\boldsymbol{\theta} \in \mathcal{X}} \|\boldsymbol{\theta} - \boldsymbol{\theta}_2\|$ then we have $\left\| \hat{\boldsymbol{\theta}}_1 - \hat{\boldsymbol{\theta}}_2 \right\| \leq \|\boldsymbol{\theta}_1 - \boldsymbol{\theta}_2\|$.*

*Proof.* Here, we only provide the proof for completeness. Combining the definition $\hat{\boldsymbol{\theta}}_1 = \min_{\boldsymbol{\theta} \in \mathcal{X}} \|\boldsymbol{\theta} - \boldsymbol{\theta}_1\|$, $\hat{\boldsymbol{\theta}}_2 = \min_{\boldsymbol{\theta} \in \mathcal{X}} \|\boldsymbol{\theta} - \boldsymbol{\theta}_2\|$ with the properties of projection operator, we have

$$\left( \hat{\boldsymbol{\theta}}_1 - \boldsymbol{\theta}_1 \right)^T \left( \hat{\boldsymbol{\theta}}_2 - \hat{\boldsymbol{\theta}}_1 \right) \geq 0 \quad \text{and} \quad \left( \hat{\boldsymbol{\theta}}_2 - \boldsymbol{\theta}_2 \right)^T \left( \hat{\boldsymbol{\theta}}_1 - \hat{\boldsymbol{\theta}}_2 \right) \geq 0.$$

From the above inequalities, we can obtain

$$\left( \hat{\boldsymbol{\theta}}_1 - \boldsymbol{\theta}_1 \right)^T \left( \hat{\boldsymbol{\theta}}_2 - \hat{\boldsymbol{\theta}}_1 \right) + \left( \boldsymbol{\theta}_2 - \hat{\boldsymbol{\theta}}_2 \right)^T \left( \hat{\boldsymbol{\theta}}_2 - \hat{\boldsymbol{\theta}}_1 \right) \geq 0 \Leftrightarrow (\boldsymbol{\theta}_2 - \boldsymbol{\theta}_1)^T \left( \hat{\boldsymbol{\theta}}_2 - \hat{\boldsymbol{\theta}}_1 \right) \geq \left\| \hat{\boldsymbol{\theta}}_2 - \hat{\boldsymbol{\theta}}_1 \right\|^2.$$

With the property that $\left\| (\boldsymbol{\theta}_2 - \boldsymbol{\theta}_1) - \left( \hat{\boldsymbol{\theta}}_2 - \hat{\boldsymbol{\theta}}_1 \right) \right\|^2 \geq 0$, we have

$$\|\boldsymbol{\theta}_2 - \boldsymbol{\theta}_1\|^2 \geq - \left\| \hat{\boldsymbol{\theta}}_2 - \hat{\boldsymbol{\theta}}_1 \right\|^2 + 2 (\boldsymbol{\theta}_2 - \boldsymbol{\theta}_1)^T \left( \hat{\boldsymbol{\theta}}_2 - \hat{\boldsymbol{\theta}}_1 \right) \geq \left\| \hat{\boldsymbol{\theta}}_2 - \hat{\boldsymbol{\theta}}_1 \right\|^2.$$

Thus, the proof has been completed. $\qquad\square$

**Lemma A.2.** *In the Algorithm 1, we have $\|\hat{\mathbf{m}}_t\| \leq 2 \|\mathbf{g}_t\|$ at each iteration $t$ if the momentum parameter satisfies $\beta_{1,t} \leq 1$.*

*Proof.* With the definition of $\hat{\mathbf{m}}_t$, we have

$$\|\hat{\mathbf{m}}_t\|^2 = \left\| \mathbf{g}_t + \frac{\beta_{1,t} \|\mathbf{g}_t\|}{\|\hat{\mathbf{m}}_{t-1}\| + \epsilon} \hat{\mathbf{m}}_{t-1} \right\|^2$$

$$\leq \|\mathbf{g}_t\|^2 + \beta_{1,t}^2 \|\mathbf{g}_t\|^2 + \frac{2\beta_{1,t} \|\mathbf{g}_t\|}{\|\hat{\mathbf{m}}_{t-1}\|} \cdot \|\mathbf{g}_t\| \cdot \|\hat{\mathbf{m}}_{t-1}\|$$

$$= (1 + \beta_{1,t})^2 \|\mathbf{g}_t\|^2.$$

With the condition that $\beta_{1,t} \leq 1$, we have $\|\hat{\mathbf{m}}_t\| \leq 2 \|\mathbf{g}_t\|$. $\qquad\square$

## A.2 PROOF OF THEOREM 4.2

*Proof.* Consider the update rule of Algorithm 1, parameters $\boldsymbol{\theta}^*$ and $\boldsymbol{\theta}_t$ satisfy

$$\Pi_{\mathcal{X}} (\boldsymbol{\theta}^*) = \boldsymbol{\theta}^* \quad \text{and} \quad \Pi_{\mathcal{X}} (\boldsymbol{\theta}_t - \alpha_t \hat{\mathbf{m}}_t) = \boldsymbol{\theta}_{t+1}.$$

With Lemma A.1, we have

$$\begin{aligned} \|\boldsymbol{\theta}_{t+1} - \boldsymbol{\theta}^*\|^2 &\leq \|\boldsymbol{\theta}_t - \alpha_t \hat{\mathbf{m}}_t - \boldsymbol{\theta}^*\|^2 \\ &= \|\boldsymbol{\theta}_t - \boldsymbol{\theta}^*\|^2 - 2\alpha_t \hat{\mathbf{m}}_t^T (\boldsymbol{\theta}_t - \boldsymbol{\theta}^*) + \alpha_t^2 \|\hat{\mathbf{m}}_t\|^2 \\ &\overset{\textcircled{1}}{=} \|\boldsymbol{\theta}_t - \boldsymbol{\theta}^*\|^2 + \alpha_t^2 \|\hat{\mathbf{m}}_t\|^2 - 2\alpha_t \left( \mathbf{g}_t + \frac{\beta_{1,t} \|\mathbf{g}_t\|}{\|\hat{\mathbf{m}}_{t-1}\| + \epsilon} \hat{\mathbf{m}}_{t-1} \right)^T (\boldsymbol{\theta}_t - \boldsymbol{\theta}^*), \end{aligned} \tag{10}$$

where $\textcircled{1}$ follows from the definition of $\hat{\mathbf{m}}_t$. Rearrange the items in Equation (10), we obtain

$$\begin{aligned} \mathbf{g}_t^T (\boldsymbol{\theta}_t - \boldsymbol{\theta}^*) \leq & \frac{1}{2\alpha_t} \left( \|\boldsymbol{\theta}_t - \boldsymbol{\theta}^*\|^2 - \|\boldsymbol{\theta}_{t+1} - \boldsymbol{\theta}^*\|^2 \right) + \frac{\alpha_t}{2} \|\hat{\mathbf{m}}_t\|^2 - \frac{\beta_{1,t} \|\mathbf{g}_t\|}{\|\hat{\mathbf{m}}_{t-1}\| + \epsilon} \hat{\mathbf{m}}_{t-1}^T (\boldsymbol{\theta}_t - \boldsymbol{\theta}^*) \\ & \overset{\textcircled{1}}{\leq} \frac{1}{2\alpha_t} \left( \|\boldsymbol{\theta}_t - \boldsymbol{\theta}^*\|^2 - \|\boldsymbol{\theta}_{t+1} - \boldsymbol{\theta}^*\|^2 \right) + \frac{\alpha_t}{2} \|\hat{\mathbf{m}}_t\|^2 + \beta_{1,t} \left[ \frac{\|\boldsymbol{\theta}_t - \boldsymbol{\theta}^*\|^2}{2\alpha_t} + \frac{\alpha_t \|\mathbf{g}_t\|^2 \|\hat{\mathbf{m}}_{t-1}\|^2}{2 (\|\hat{\mathbf{m}}_{t-1}\| + \epsilon)^2} \right] \\ & \leq \frac{1}{2\alpha_t} \left( \|\boldsymbol{\theta}_t - \boldsymbol{\theta}^*\|^2 - \|\boldsymbol{\theta}_{t+1} - \boldsymbol{\theta}^*\|^2 \right) + \frac{\alpha_t}{2} \|\hat{\mathbf{m}}_t\|^2 + \beta_{1,t} \left( \frac{\|\boldsymbol{\theta}_t - \boldsymbol{\theta}^*\|^2}{2\alpha_t} + \frac{\alpha_t \|\mathbf{g}_t\|^2}{2} \right), \end{aligned}$$

where ① holds by the Young's inequality. By Assumption 4.1, with the convexity of all $f_i$, we have

$$\sum_{t=1}^{T} [f_t(\boldsymbol{\theta}_t) - f_t(\boldsymbol{\theta}^*)] \leq \sum_{t=1}^{T} \mathbf{g}_t^T (\boldsymbol{\theta}_t - \boldsymbol{\theta}^*).$$

Submit Equation (A.2) to the above inequalities, we obtain

$$\sum_{t=1}^{T} [f_t(\boldsymbol{\theta}_t) - f_t(\boldsymbol{\theta}^*)] \leq \underbrace{\sum_{t=1}^{T} \sum_{i=1}^{d} \frac{1}{2\alpha_t} \left[ (\theta_{t,i} - \theta_i^*)^2 - (\theta_{t+1,i} - \theta_i^*)^2 \right]}_{T_1} +$$

$$\underbrace{\sum_{t=1}^{T} \sum_{i=1}^{d} \frac{\alpha_t}{2} \hat{m}_{t,i}^2}_{T_2} + \underbrace{\sum_{t=1}^{T} \sum_{i=1}^{d} \frac{\alpha_t \beta_{1,t}}{2} g_{t,i}^2}_{T_3} + \underbrace{\sum_{t=1}^{T} \sum_{i=1}^{d} \frac{\beta_{1,t}}{2\alpha_t} (\theta_{t,i} - \theta_i^*)^2}_{T_4}, \tag{11}$$

where $\beta_{1,t} = \beta \lambda^{t-1}$ is monotonically decreasing with $t$. For $T_1$, we have

$$\frac{1}{2} \sum_{t=1}^{T} \sum_{i=1}^{d} \frac{1}{\alpha_t} \left[ (\theta_{t,i} - \theta_i^*)^2 - (\theta_{t+1,i} - \theta_i^*)^2 \right]$$

$$= \frac{1}{2} \left[ \frac{1}{\alpha_1} \sum_{i=1}^{d} (\theta_{1,i} - \theta_i^*)^2 + \sum_{t=2}^{T} \sum_{i=1}^{d} \left( \frac{1}{\alpha_t} - \frac{1}{\alpha_{t-1}} \right) (\theta_{t,i} - \theta_i^*)^2 \right]$$

$$\leq \frac{D_\infty^2}{2} \left[ \sum_{i=1}^{d} \frac{1}{\alpha_1} + \sum_{t=2}^{T} \sum_{i=1}^{d} \left( \frac{1}{\alpha_t} - \frac{1}{\alpha_{t-1}} \right) \right] \tag{12}$$

$$= \frac{D_\infty^2}{2} \sum_{i=1}^{d} \frac{1}{\alpha_T} = \frac{D_\infty^2}{2\alpha} \sqrt{dT},$$

where we set $\alpha_t = \alpha \sqrt{d/t}$. For $T_2$ in Equation (11), we obtain

$$\frac{1}{2} \sum_{t=1}^{T} \sum_{i=1}^{d} \alpha_t \hat{m}_{t,i}^2 = \frac{1}{2} \left[ \sum_{t=1}^{T-1} \sum_{i=1}^{d} \alpha_t \hat{m}_{t,i}^2 + \sum_{i=1}^{d} \alpha_T \hat{m}_{T,i}^2 \right]$$

$$\overset{①}{\leq} \frac{1}{2} \left[ \sum_{t=1}^{T-1} \sum_{i=1}^{d} \alpha_t \hat{m}_{t,i}^2 + \frac{4\alpha \sqrt{d}}{\sqrt{T}} \sum_{i=1}^{d} g_{T,i}^2 \right] \leq \frac{1}{2} \left[ \sum_{t=1}^{T-1} \sum_{i=1}^{d} \alpha_t \hat{m}_{t,i}^2 + \frac{4 G_\infty \alpha \sqrt{d}}{\sqrt{T}} \sum_{i=1}^{d} |g_{T,i}| \right]$$

$$= 2 G_\infty \alpha \sqrt{d} \sum_{t=1}^{T} \frac{1}{\sqrt{t}} \sum_{i=1}^{d} |g_{t,i}| = 2 G_\infty \alpha \sqrt{d} \sum_{i=1}^{d} \sum_{t=1}^{T} \frac{|g_{t,i}|}{\sqrt{t}}$$

$$\overset{②}{\leq} 2 G_\infty \alpha \sqrt{d} \sum_{i=1}^{d} \|g_{1:T,i}\|_2 \sqrt{\sum_{t=1}^{T} \frac{1}{t}} \leq 2 G_\infty \alpha \sqrt{(1 + \log T) d} \sum_{i=1}^{d} \|g_{1:T,i}\|_2, \tag{13}$$

where we introduce Lemma A.2 in inequality ①. Besides, we get ② through Cauchy-Schwarz inequality. With similar proof techniques, we have $T_3$ satisfied

$$\sum_{t=1}^{T} \sum_{i=1}^{d} \frac{\alpha_t \beta_{1,t}}{2} g_{t,i}^2 \leq \sum_{t=1}^{T-1} \sum_{i=1}^{d} \frac{\alpha_t \beta_{1,t}}{2} g_{t,i}^2 + \frac{\alpha \beta \sqrt{d}}{2\sqrt{T}} \sum_{i=1}^{d} g_{T,i}^2 \leq \sum_{t=1}^{T-1} \sum_{i=1}^{d} \frac{\alpha_t \beta_{1,t}}{2} g_{t,i}^2 + \frac{G_\infty \alpha \beta \sqrt{d}}{2\sqrt{T}} \sum_{i=1}^{d} |g_{T,i}|$$

$$= \frac{G_\infty \alpha \beta \sqrt{d}}{2} \sum_{i=1}^{T} \frac{1}{\sqrt{t}} \sum_{i=1}^{d} |g_{t,i}| = \frac{G_\infty \alpha \beta \sqrt{d}}{2} \sum_{i=1}^{d} \sum_{t=1}^{T} \frac{|g_{t,i}|}{\sqrt{t}}$$

$$\leq \frac{G_\infty \alpha \beta \sqrt{d}}{2} \sum_{i=1}^{d} \|g_{1:T,i}\|_2 \sqrt{\sum_{t=1}^{T} \frac{1}{t}} \leq \frac{G_\infty \alpha \beta}{2} \sqrt{(1 + \log T) d} \sum_{i=1}^{d} \|g_{1:T,i}\|_2. \tag{14}$$

Since Equation (14) is sufficient for our regret convergence analysis, we have not tried to optimize it further. Then, for $T_4$, we have

$$
\begin{aligned}
\sum_{t=1}^{T} \sum_{i=1}^{d} \frac{\beta_{1,t}}{2\alpha_t} \left(\theta_{t,i} - \theta_i^*\right)^2 &= \sum_{t=1}^{T} \frac{\beta_{1,t}}{2\alpha_t} \sum_{i=1}^{d} \left(\theta_{t,i} - \theta_i^*\right)^2 \overset{\text{\textcircled{1}}}{=} \frac{\beta_{1,t}}{2\alpha\sqrt{d}} \sum_{t=1}^{T} \sqrt{t}\lambda^{t-1} \sum_{i=1}^{d} \left(\theta_{t,i} - \theta_i^*\right)^2 \\
&\leq \frac{\sqrt{d}\beta D_\infty^2}{2\alpha} \sum_{t=1}^{T} \sqrt{t}\lambda^{t-1} \leq \frac{\sqrt{d}\beta D_\infty^2}{2\alpha} \sum_{t=1}^{T} t\lambda^{t-1} \overset{\text{\textcircled{2}}}{\leq} \frac{\sqrt{d}\beta D_\infty^2}{2\alpha\left(1-\lambda\right)^2},
\end{aligned}
\tag{15}
$$

where \textcircled{1} follows from the definition of $\beta_{1,t}$ and $\alpha_t$, and \textcircled{2} establishes because of the geometric series summation rule. As a result, submitting Equation (12), Equation (13), Equation (14) and Equation (15) back into Equation (11), we have

$$
R(T) \leq \frac{D_\infty^2}{2\alpha}\sqrt{dT} + 3G_\infty\alpha\sqrt{(1+\log T)\,d} \sum_{i=1}^{d} \|g_{1:T,i}\|_2 + \frac{\sqrt{d}\beta D_\infty^2}{2\alpha\left(1-\lambda\right)^2}.
\tag{16}
$$

This completes the proof. $\qquad\square$

### A.3 Proof of Corollary 4.3

*Proof.* From Theorem 4.2, we have

$$
R(T) \leq \frac{D_\infty^2}{2\alpha}\sqrt{dT} + 3G_\infty\alpha\sqrt{(1+\log T)\,d} \sum_{i=1}^{d} \|g_{1:T,i}\|_2 + \frac{\sqrt{d}\beta D_\infty^2}{2\alpha\left(1-\lambda\right)^2}.
\tag{17}
$$

For the second term, we have

$$
\sum_{i=1}^{d} \|g_{1:T,i}\|_2 \leq \sum_{i=1}^{d} \sqrt{\sum_{t=1}^{T} g_{t,i}^2} \leq dG_\infty\sqrt{T}
\tag{18}
$$

Combining the above inequality with Equation (17), we obtain

$$
R(T) = \tilde{O}(\sqrt{T})
$$

$\qquad\square$

