# OpenReview forum: "Acutum: When Generalization Meets Adaptability"
_ICLR.cc/2020/Conference — Reject_

### Official Review · AnonReviewer2 · 2019-10-23
**Official Blind Review #2**

**Rating:** 3

**Review:**

This paper proposes new stochastic optimization methods to achieve a fast convergence of adaptive SGD and preserve the generalization ability of SGD. The idea is to let the search direction opposite to the gradient at the current batch of examples and a bit orthogonal to previous batch of examples. The algorithm is easy to implement and backed up with regret bounds. Several experimental results are also reported to verify the effectiveness of the algorithm.

The regret bound in Theorem 4.2. is not quite satisfactory. For example, Adagrad in Duchi et al (2011) can achieve the regret bound $O(\sum_{i=1}^{d}\|g_{1:T,i}\|_2)$. However, there is an additional factor of $\sqrt{d}$ in the regret boun in Theorem 4.2, which is not appealing in high dimensional problems. The regret analysis follows largely from standard arguments.

I can not follow the last identity of (10). It should not hold since there is a missing $\epsilon$ there.

In both eq (14) and (15), it is not clear to me why the authors divided the summation into two parts. Also, $\sqrt{\sum_{t=1}^{T}\frac{1}{\sqrt{t}}}$ should be $\sqrt{\sum_{t=1}^{T}\frac{1}{t}}$ there.

----------------------
After rebuttal:

I have read the author response. Unlike AdaGrad, the proposed algorithm does not show an advantage over standard OGD. I would like to keep my original score.

**Experience Assessment:**

I have published one or two papers in this area.

**Review Assessment: Checking Correctness Of Derivations And Theory:**

I assessed the sensibility of the derivations and theory.

**Review Assessment: Checking Correctness Of Experiments:**

I did not assess the experiments.

**Review Assessment: Thoroughness In Paper Reading:**

I read the paper at least twice and used my best judgement in assessing the paper.

---

> ### Author Response · Authors · 2019-11-08
> **Response to Review #2**
>
> Thank you for your valuable suggestions, which is indeed helpful to improve our theoretical analysis. Such improvements can answer your concerns straightforwardly.
>
> Actually, "let the search direction opposite to the gradient at the current batch of examples and a bit orthogonal to the previous batch of examples" is our observation inspired by the sub-problem of Adagrad. Based on such an observation, the intuition of our Acutum is to find a descent direction that can both make an acute angle with both the current batch-gradient and the approximate descent direction of previous batch loss. In practice, choosing the direction as the angle bisector of $g_t$ and $\hat{m}_t$ on the hyper-plane they form can make the experimental results satisfied.
>
> Q: The regret bound in Theorem 4.2. is not quite satisfactory...
> A: Actually, we find the additional factor of $\sqrt{d}$ in the regret bound in Theorem 4.2 comes from our setting of step size — $\alpha_t = \alpha\sqrt{d/t}$. The additional factor of $\sqrt{d}$ can be removed as long as we set $\alpha_t = \alpha/\sqrt{t}$. We have updated our regret proof in the latest version of our paper and appendix where you can find more details.
>
> Q: I can not follow the last identity of (10)...
> A: For the missing $\epsilon$ in Eq.(10), we previously thought of it as a small amount that maintains numerical stability (preventing the denominator from being 0). Therefore we ignored the $\epsilon$ in our analysis. We have taken $\epsilon$ into account in the latest version of our paper, and we can get the same regret bound with a trivial calculation.
>
> Q: In both eq (14) and (15) ...
> A: Eq.(14) and Eq.(15) are different parts of Eq.(11), we divide the summation into two parts for a more clear statement. We have fixed the typos about $\sqrt{\sum_{t=1}^t1/\sqrt{t}}$ in our latest version.
>
> Lastly, please allow us to emphasize the importance of this article. We are the first to explain the effect of second moments through constructing approximate subproblems and point that "second moments essentially penalize the projection of the current descent direction on the exponential moving average of previous gradients", which is totally different from most of the previous works. We believe such consideration could be instructive to the design of future optimizers. Furthermore, our Acutum only utilizes ONLY the first moments to achieve a comparable efficiency with Adam-type optimizers and even a better generalization, which illustrates that our intuition is effective from a practice perspective. Theoretically, our Acutum enjoys a regret of $\tilde{O}(1/\sqrt{T})$ which is the same as the best of known bound for general convex online learning problems (Adam, AMSGrad, etc).
>
> We hope you find your concerns satisfactorily addressed by our response, which has also been reflected in the revised paper. Please let us know if you have more comments or any other suggestions.

---

> > ### Comment · AnonReviewer2 · 2019-11-14
> > **response**
> >
> > Thank you for your response to my comments. Although the factor $\sqrt{d}$ can be removed in front of $\sum_{i=1}^d\|g_{1:T},i\|_2$, the regret bound involves $d\sqrt{T}$ in the first term. In this sense, the regret bound admits no advantages over the online gradient descent.
> >
> > A minor issue in the proof of Lemma A.2: the first inequality generally not holds since $g_t^T\hat{m}_{t-1}$ may be negative. However, this can be simply addressed if the authors adds absolute value there.

---

> > > ### Author Response · Authors · 2019-11-15
> > > **Response to Review #2**
> > >
> > > Thank you for your response! Actually, we can only obtain an $O(d\sqrt{T})$ upper bound for the term $\sum_{i=1}^d\left\|g_{1:T,i}\right\|_2$ in all Adam-type optimizers, which is just like Eq.(18) in our paper. With such upper bound, choosing step size as $\alpha/\sqrt{t}$ can indeed balance the order of $d$ among different terms in Eq.(9) (the order of $d$ of the first term and the second term in Eq.(9) are both $O(d)$). However, it also makes the regret admit no advantages over the online gradient descent, even for other Adam-type optimizers. Thus, if we choose the step size as $O(\alpha\sqrt{d/t})$, we can guarantee the same regret $O(\sqrt{dT})$ with SGD under the condition $\sum_{i=1}^d\left\|g_{1:T,i}\right\|_2\le O(\sqrt{T})$. In our opinion, providing the convex regret bound is to qualitatively show that Acutum is convergent. As for the analysis of the convergence rate, we may utilize standard frameworks, say[1] or [2]. As you know, new ideas are required to derive tighter arguments. From a practical perspective, we believe our experiments can clearly show the convergence rate and the potential of Acutum. Besides, we didn't mention that improving the regret bound as our contribution. It's worth to mention that our main contribution is to propose a new insight to explain all the other Adam-type methods, and then demonstrate the improvement by following the novel understanding.
> > >
> > > For your second question, we will add the absolute value sign there in our next revision. Thanks a lot for your carefully proof-checking:)
> > >
> > > [1]Chen, Xiangyi, et al. "On the convergence of a class of adam-type algorithms for non-convex optimization." arXiv preprint arXiv:1808.02941 (2018).
> > > [2]Zaheer, Manzil, et al. "Adaptive methods for nonconvex optimization." Advances in Neural Information Processing Systems. 2018.

---

### Official Review · AnonReviewer1 · 2019-10-23
**Official Blind Review #1**

**Rating:** 1

**Review:**

This paper attempts to remove the use of the second moment in Adam in order to improve the generalization ability of Adam.
- Apriori, it is not clear why removing the second moment is important. Does it improve the generalization or decrease the runtime substantially?
- Please clarify how our method compares against Yogi (Adaptive Methods for Nonconvex Optimization, Zaheer' 2018) and the more recent RADAM (ON THE VARIANCE OF THE ADAPTIVE LEARNING
RATE AND BEYOND, Liu 2019) , both theoretically and empirically.
- The paper is meandering and confusing. State your update/algorithm and then explain how it compares against the other methods. Besides, there might be technical problems with it.

See the detailed review below:
- Section 1: "the generalization results (of adaptive methods) cannot be as good as SGD". Please cite the relevant papers, (for example, Wilson. 2017)  that show this empirically.
-  Section 1: "the proposed algorithm outperforms Adam in convergence speed" - Please clarify what "convergence speed" refers to. Is it the number of gradient evaluations, the rate of convergence or the wall-clock time. Please state how did you conclude this.
- Section 2: The update rule of Adagrad is incorrect. The step-size is constant \alpha and it is decreased over time because of the v_t term.
- Section 3: There is no guarantee on the approximation in Equation 5. Young's inequality and the resulting upper bound can be quite loose.
- Section 3: In general, it is not possible to have an update that decreases the loss on the current batch, but does not increase the previous batches loss. It is always possible to construct a counter-example to this.
- "In practice, the computational cost of computing" the gradient for all i is expensive. Indeed, this is batch gradient descent. I am not sure how this is relevant to the discussion in the paper.
- Cite and compare against the variance reduced methods (Stochastic Average Gradient, Schmidt, 2013; SVRG, Johnson, 2013) as these try to "approximate" the full gradient in order to decrease the variance.
- The derivation/formulation of Equation 8 is not clear to me. Why is the \hat{m} normalized?
- In algorithm 1, it seems you need to choose the sequence of step-sizes \alpha_t and \beta_t. How is this an adaptive method then? How are these sequences chosen theoretically and practically? Please clarify this.
- Section 4: Please compare the resulting regret bound to that of Adam, Adagrad and AMSgrad. Why does \alpha_t = O(1/t)? If we have to decrease the step-size according to a sequence, why should I not use standard SGD?
- Section 5: "we decay the learning rate by 0.1 every 50 epochs" This is not aligned with either the theory or the algorithm you proposed.


**Experience Assessment:**

I have published one or two papers in this area.

**Review Assessment: Checking Correctness Of Derivations And Theory:**

I carefully checked the derivations and theory.

**Review Assessment: Checking Correctness Of Experiments:**

I carefully checked the experiments.

**Review Assessment: Thoroughness In Paper Reading:**

I read the paper thoroughly.

---

> ### Author Response · Authors · 2019-11-08
> **Response to Review #1**
>
> Thanks for your helpful suggestions and comments. We address your concerns as follows.
>
> Q: Motivation for removing the second moment...
> A: The motivation for removing the second moment is both to decrease the wall-clock time and to improve the generalization. For the wall-clock time, Acutum finds a better descent direction inspired by the sub-problem of Adagrad. For the generalization, we have the following observation --- The generalization of adaptive methods (with second moments) cannot be as good as SGD with ONLY first moments.
>
> Q: Comparing against Yogi...
> A: We have added extra evaluations of Yogi and Radam in our new version. Theoretically, we have proved that Acutum enjoys a regret of $\tilde{O}(1/\sqrt{T})$ which is the same as the best known bound for convex online learning problems. One of our contributions is to illustrate that a prototype optimizer Acutum is comparable to existing Adam-type optimizers.
>
> Q: The paper is confusing...
> A: The update procedure and alg. are shown in Eq 8 and Alg 1 clearly. We have compared with Acutum, Adagrad, and RMSProp in Section 3. Hence, Acutum is a totally new method, *NOT* a minor increments/variants by removing the second moments.
>
> Q: convergence speed...
> A: We have replaced the "convergence speed" with "wall-clock time".
>
> Q: The update rule of Adagrad is incorrect...
> A: From Thm 5 of Adagrad, we can obtain that if Adagrad requires an $\tilde{O}(1/\sqrt{T})$ regret convergence in convex settings, it still needs a step size with order $O(1/\sqrt{t})$. Adagrad accepts a constant step size both in the alg. description and in the practice, though.
>
> Q: No guarantee in Eq 5...
> A: Here, we only utilize Young's inequality to investigate the properties of the optimal solution of the Adagrad sub-problem by constructing an upper bound objective, but not analysis the theory in this part. Furthermore, based on such properties, we introduce the intuition of our Acutum.
>
> Q: Update decreases the loss on the current batch,...
> A: We agree that it is possible to construct a counter-example to prevent us to find a direction that can descent all the batch loss. To prevent the previous batch loss from increasing, the sub-problem of Adagrad introduces a regularization $\sum\|(\theta- \theta_t)^Tg_\tau\|^2$ rather than some strict constraints. It somehow acts like the soft margin SVM.
>
> Q: Computing the gradient for all i is expensive...
> A: Here we want to illustrate that there is a gap between the physical intuition (penalize the projection of current descent direction on previous gradients) and the formulation of Adagrad. What they really want to calculate $\nabla f_{A_i}(\theta_t)$ is the expensive and be approximated by $\nabla f_{A_i}(\theta_i)$.
>
> Q: For variance reduced methods...
> A: We consider the optimization in the online learning setting rather than the stochastic optimization settings. Thus, we neither approximate the full gradient nor have any operation related to variance reduction. As you may know, it is a common setting for the theoretical analysis of Adam-type optimizers (Adam, Adagrad, Amsgrad).
>
> Q: Eq 8 is not clear...
> A: In Section 3, we normalized $\hat{m}_t$ to make the current descent direction $\hat{m}_{t+1}$ and current batch gradient $g_t$ an acute angle. One can easily see that it makes $\hat{m}_{t+1}$ and $\hat{m}_t$ an acute angle. You can consider $\hat{m}_{t+1}$ as the angle bisector of $g_t$ and $\hat{m}_t$ on the hyper-plane they form. We strongly believe it is a novel observation and one may improve by following our insight.
>
> Q: Choose the sequence of step-sizes...
> A: As you know, adaptivity means the process of updating parameters also depends on the history and current situation. We utilized the norm of $\hat{m}_t$ to adaptively adjust the update direction. In practice, we set $\alpha_t$ as some constant, and $\beta_t=1$ as shown in the paragraph after the alg. In theory, to guarantee the regret $\tilde{O}(1/\sqrt{T})$, we set $\alpha_t=O(1/\sqrt{t})$ and $\beta_t=\beta \lambda^{t-1}$ where $\beta, \lambda \in (0,1)$, which has been shown in Thm 4.2.
>
> Q: Compare the regret bound to others
> A: To guarantee the regret $\tilde{O}(1/\sqrt{T})$, most of Adam-type optimizers require an $O(1/\sqrt{t})$ step size, e.g., Thm 5 of Adagrad, Thm 4.1 of Adam and Thm 4 of AMSGrad. In practice, when we set a proper constant step size, Adagrad, Adam, and AMSGrad can also work. This phenomenon also appears in Acutum.
>
> Q: decay the lr by 0.1...
> A: In the CV community, lr decay is a common trick of getting a better result. To achieve the SOTA testing accuracy, we did such work for all optimizers. Such settings also appear in related work, e.g., Padam.
>
> We have revised our paper accordingly. Almost all existing adaptive methods require second moments. In a novel point of view, we achieved similar results with ONLY first moments. We also proved our conclusion both theoretically and experimentally in our paper.
> Any further comments on the paper are more than welcome.

---

### Official Review · AnonReviewer3 · 2019-10-24
**Official Blind Review #3**

**Rating:** 6

**Review:**

Balancing the generalization and convergence speed is an important direction in deep learning. This paper propose a new method to balance them. That is very interesting to me. However, I have several concerns as follows.
1. I cannot agree that "fewer hyper-parameters" in Algorithm 1. The authors should provide more materials to support this claim, such as a comparison table of different mathods.
2. In Theorem 42, how to set the parameter \epsilon to obtain the conclusion.
3. The authors should provide more comparison of the theoretical results of different SGD algorithms (such Adam, RMSProp ...)
Some minor issues:
1. The conditions of formulation of Eq. (6) should be placed together with Eq. (6).
2. What is the definition to c_2,0, c_2,1 .... It is not clear to readers.
3. The presentation and structure of the paper should be improved further, such as adding several subsection in Sec. 3, so that readers could easily follow the main idea of this paper.

**Experience Assessment:**

I have published one or two papers in this area.

**Review Assessment: Checking Correctness Of Derivations And Theory:**

I assessed the sensibility of the derivations and theory.

**Review Assessment: Checking Correctness Of Experiments:**

I assessed the sensibility of the experiments.

**Review Assessment: Thoroughness In Paper Reading:**

I read the paper thoroughly.

---

> ### Author Response · Authors · 2019-11-08
> **Response to Review #3**
>
> Thank you for your supportive review and useful suggestions.
>
> Q: I cannot agree that ``fewer hyper-parameters'' in Algorithm 1...
> A: For the claim about ``fewer hyper-parameters'', most of Adam-type optimizers require the hyper-parameters $\alpha$, $\beta_{1}, \beta_{2}$ corresponding to the step size, the coefficients of the first moments and the second moments. Besides, SGD momentum also requires the hyper-parameters tuning about $\alpha$ and $\beta$ which are denoted as the step size and coefficients of momentum, respectively. In general cases, when we consider $\hat{m}_t$ as the angle bisector of $g_t$ and $\hat{m}_t$ on the hyperplane they form, we can set $\beta_{1,t} = 1$ in Acutum. That is to say, the only hyper-parameter we need to adjust is $\alpha$ — the step size. Therefore, we think Acutum requires fewer hyper-parameters.
>
> Q: In Theorem 4.2, how to set the parameter...
> A: Actually, we introduce the $\epsilon$ only for preventing the denominator from being 0. In the latest version of Theorem 4.2, we have illustrated that for any positive $\epsilon$, we can maintain the convex regret bound. From a practice perspective, we hope the $\epsilon$ as small as possible because a large $\epsilon$ will make the angle among $\hat{m}_t$, $\hat{m}_{t-1}$ and $g_t$ unstable.
>
> Q: The authors should provide more comparison of the theoretical results of different SGD algorithms...
> A: Thank you for your suggestion, Acutum enjoys a regret of $\tilde{O}(1/\sqrt{T})$ which is the same as the best of known bound for general convex online learning problems (e.g., Adam, AMSGrad and Adagrad).
>
> Q: The conditions of the formulation of Eq. (6) should be placed together with Eq. (6).
> A: Thank you for your suggestion. Your suggestion has been updated in our new version.
>
> Q: What is the definition to $c_{2,0}, c_{2,1}$ .... It is not clear to readers.
> A: In Eq.(7), we only want to clarify our intuition. The constants $c_{2,0}, c_{2,1}$ are only some numbers independent from the variables we need to optimize. In this section, we just show the intuition and math forms for the final objectives, not the exact resulting functions. We are sorry for the confusion.
>
> Q: The presentation and structure of the paper should be improved further ...
> A: Thanks for your suggestion! Convincing readers is the most important work for research papers. We are carefully considering how to add subsection in Section 3 to make readers follow the main idea of this paper more easily, and we will update the relevant context in our next version:)
>
> The response above has been reflected in our revised paper accordingly. Please let us know if you have any further suggestions.

---

### Decision · Program_Chairs · 2019-12-19

**Decision:**

Reject

**Comment:**

The paper addresses an important problem of finding a good trade-off between generalization and convergence speed of stochastic gradient methods for training deep nets. However, there is a consensus among the reviewers, even after rebuttals provided by the authors, that  the contribution is somewhat limited and the paper may require additional work before it is ready to be published.